# Aberrant Corneal Homeostasis in Neurosurgery-Induced Neurotrophic Keratopathy

**DOI:** 10.3390/jcm11133804

**Published:** 2022-06-30

**Authors:** Shimpei Oba, Kaoru Araki-Sasaki, Tomoyuki Chihara, Takashi Kojima, Dogru Murat, Kanji Takahashi

**Affiliations:** 1Department of Ophthalmology, Kansai Medical University, 2-5-1 Shinmachi, Hirakata, Osaka 573-1010, Japan; kansai.med@gmail.com (S.O.); mail4chiharat@gmail.com (T.C.); takahask@takii.kmu.ac.jp (K.T.); 2Department of Ophthalmology, Keio University School of Medicine, 2-15-45 Mita, Minato-ku, Tokyo 108-8345, Japan; tkojkoj@mac.com (T.K.); muratodooru2005@yahoo.co.jp (D.M.)

**Keywords:** neurotrophic keratopathy, trigeminal nerve, corneal basal cells, dendritic cells Schirmer test

## Abstract

The characteristic features of neurotrophic keratopathy have been well documented by in vivo and in vitro studies using animal models. However, case reports of neurotrophic keratopathy induced by neurosurgery are limited. We describe the clinical characteristics, anterior segment optical coherence tomography (AS-OCT) and in vivo confocal microscopy (IVCM) findings of neurotrophic keratopathy induced by surgery for intracranial lesions. This is a case series including 6 eyes of 3 patients (mean age, 69.67 ± 12.50 years) with unilateral neurotrophic keratopathy. The clinical findings of three patients were described and IVCM findings of three patients were analyzed. The duration of neuropathy ranged from 2 to 30 years (median, 22 years). Thickening of the epithelial layer and higher reflection density of the anterior stroma were observed during the healing process using AS-OCT. The mean nerve fiber density of the subepithelial plexus, as determined by IVCM, was 1943 ± 1000 μm/mm^2^ for neurotrophic eyes and 2242 ± 600.3 μm/mm^2^ for contralateral eyes (*p* = 0.0347). The mean respective dendritic cell densities were 30.8 ± 21.8 and 6.25 ± 5.59 cells/mm^2^ (*p* < 0.0001), while the mean basal cell sizes were 259 ± 86.5 and 185 ± 45.9 μm^2^ (*p* < 0.0001), respectively. These findings suggest that neurosurgery-induced neurotrophic keratopathy may be associated with alterations in the healing process and immune cell distribution in the cornea.

## 1. Introduction

Neurotrophic keratopathy is a disease in which the corneal epithelial cells cannot regenerate properly due to impairment of the trigeminal nerve caused by intracranial lesions, surgery, diabetes mellitus, or corneal herpes. The main diagnostic criterium of NK is hypesthesia or anesthesia of the cornea.

The trigeminal nerve is believed to be responsible for the homeostasis of the corneal epithelium. Delayed wound healing, decreased cell proliferation, weak intercellular adhesion, and excessive inflammation have been reported in animal models of neurotrophic keratopathy [1,2,3,4,5].

In addition, a relationship between neurotransmitters in the trigeminal nerve and immune cells has long been suggested [6,7,8]. Recently, the concept of neuro-immune linkage has been established: studies of animal models have revealed that dendritic cells and nerve fibers are in close contact, and that damage to the trigeminal nerve alter the distribution of dendritic cells in the cornea [9,10,11].

Although the pathogenesis of neurotrophic keratopathy has been analyzed using animal models to some extent, human cases have been reported mainly with neurotrophic keratopathy caused by corneal herpes, ocular infections, diabetes mellitus, and after ocular surgery [12,13,14,15]. However, the pathophysiology of these cases may have been modified by microbial insults, surgical injury, and metabolic abnormalities, and thus are not suitable for analyzing pure corneal changes due to neuropathy. It would be desirable to elucidate the pathophysiological changes in unilateral trigeminal nerve neuropathy caused by intracranial lesions because the differences between both eyes could be compared in each patient. This can eliminate the influence of individual differences. However, only a few clinical cases of neurotrophic keratopathy caused by intracranial lesions have been reported to date [16,17].

We investigated the changes of corneal findings in neurosurgery-induced unilateral neurotrophic keratopathy by comparing the affected and contralateral eyes.

## 2. Materials and Methods

This is a case series of 6 eyes of 3 patients with neurotrophic keratopathy. All cases suffered from unilateral neurotrophic keratopathy induced by surgery for intracranial lesions. The classification of neurotrophic keratopathy was based on Mackie’s classification [18].

### 2.1. Institutional Review and Recruitment

The procedures used were approved by the Institutional Review Board of Kansai Medical University (No. 2020018) and they conformed to the tenets of the Declaration of Helsinki. A signed informed consent form was obtained from each of the patients for the examination procedures, and the use of any data for future publications. The patients were assured of anonymity.

### 2.2. Subjects and Methods

The subjects were three patients and six eyes with unilateral neurotrophic keratopathy due to neurosurgery. We performed slit-lamp microscopic examination and anterior segment optical coherence tomography (AS-OCT) imaging (The SPECTRALIS^®^ OCT2 Module with Anterior Segment Module, Heidelberg Engineering GmbH., Heidelberg, Germany).

Three cases have received the in vivo confocal microscopic (IVCM) examination (Heidelberg Retina Tomograph III with Rostock Cornea Module (HRT3/RCM), Heidelberg Engineering GmbH., Heidelberg, Germany) to measure corneal trigeminal nerve density, nerve tortuosity, corneal epithelial basal cell area, and dendritic cell density. One hundred sequences of photographs in the central region of the cornea were taken and the five images for each analysis were carefully selected. The patient was asked to fixate on the front and the image was taken at the center of the cornea. We photographed the cornea from the anterior aspect. We then selected images that were taken at a depth where the nerve plexus could be observed first and yet were appeared on the entire image.

For analysis of sub-basal nerve plexus, the five images just beneath Bowman’s layer from IVCM examinations were randomly extracted for each case, the nerve fibers in one field of a photograph were delineated using Image J software (National Institutes of Health, Bethesda, MD, USA), and the number of nerve fibers was converted into length per square millimeter. The degree of tortuosity was analyzed by the same method using Image J, according to previous reports [19]. To analyze the size of corneal epithelial basal cells, data from five images just above Bowman’s layer were randomly selected for each case, and the size of 15 consecutive basal cells in one field of view was measured using Image J. The mean area of 75 corneal epithelial basal cells per eye was calculated and analyzed. The number of dendritic cells was counted by the naked eye in one visual field (400 μm × 400 μm) using five photos by two examiners in masked fashion at the level of the subepithelial plexus from each case and given as cells per square millimeter. The evaluator of all the images was blinded about the affected eyes and the contralateral eyes.

### 2.3. Statistical Analysis

Analysis of variance (ANOVA) was used for statistical analysis. Statistical significance was established at *p* < 0.05.

## 3. Results

A list of patients is presented in Table 1. The age of the patients ranged from 52 to 79 years (mean, 69.67 ± 12.50 years), and they were all females. Neurosurgical procedures included auditory schwannoma resection in 3 patients, and post-γ knife irradiation for brain tumor in 1 patient. The postoperative follow-up ranged from 2 to 30 years, with median of 16 years. Corneal perception was measured with the Cochet–Bonnet corneal esthesiometer and was < 5 mm in all three cases. The contralateral eyes were more than 50 mm in all cases. The frequency of recurrence of corneal epithelial erosion was frequent (once a year on average) in Cases 1 and 2, while Case 3 had no history of epithelial erosion. Case 1 and 2 were classified into stage 1–2 and case 3 into stage 1 according to Mackie’s classification. There was no facial paralysis nor other cerebral neurological disease in all cases. All patients were treated with antimicrobial ointment and eye patch or therapeutic contact lenses when corneal damage occurred. When the corneal epithelium was stable, hyaluronic acid eye drops were administered.

### 3.1. Slit-Lamp Examination and Anterior OCT Observation

Figure 1 shows the slit-lamp micrographs and anterior segment OCT in our cases. OCT was performed when the corneal erosions were almost healed. Corneal opacity or erosion in the neurotrophic eye was located slightly below the pupil, and the surface of the epithelium was slightly elevated when the epithelial erosion was healed. Anterior segment OCT also showed a hyperintense reflection of the superficial epithelium (arrow) in Cases 1 and 2). Fluorescein staining showed focal areas of disordered epithelial thickening without the vortex pattern seen in the normal wound healing process. Although the neurotrophic eye of Case 3 had no corneal erosion, superficial punctate keratopathy was observed in the lower part of the cornea.

### 3.2. Nerve Fiber Density of Subepithelial Nerve Plexus

Representative photographs of nerve fibers observed by confocal laser microscopy in Cases 1, 2, and 3 are shown in Figure 2. The number of nerve fibers decreased in the neurotrophic eye (upper panel) more than in the contralateral eye (lower panel). Five photographs were randomly selected in each eye, with mean nerve fiber densities of 1943 ± 1000 μm/mm^2^ in the affected eye and 2242 ± 600.3 μm/mm^2^ in the contralateral eye. ANOVA results showed that the nerve fiber density was significantly decreased in neurotrophic eyes relative to that in contralateral eyes (Table 2). The mean tortuosity grading in neurotrophic and contralateral eyes were 1.85 and 1.78, respectively, with a significant difference (*p* = 0.0438).

### 3.3. Dendritic Cell Densities

Dendritic cells at the level of the subepithelial nerve plexus observed by confocal laser microscopy are shown in Figure 3. As shown in Figure 3a, cells with dendrites visible to the naked eye were measured. The mean number of dendritic cells was determined for each eye in each patient using five randomly selected photographs. The mean dendritic cell densities in Cases 1, 2, and 3 were 30.8 ± 21.8 cells/mm^2^ in the neurotrophic eye and 6.25 ± 5.59 cells/mm^2^ in the contralateral eye, with a significant difference (*p* < 0.0001) (Table 2).

### 3.4. Size of Corneal Epithelial Basal Cells

Figure 4 shows corneal epithelial basal cells observed by confocal laser microscopy: 15 consecutive cells in 1 photograph were depicted for analysis, and 75 cells per eye were measured and employed for analysis. The mean basal cell size was 259 ± 86.5 μm^2^ in the affected eye and 185 ± 45.9 μm^2^ in the contralateral eye. ANOVA results showed that the area of corneal epithelial basal cells was significantly increased in neurotrophic eyes relative to that in contralateral eyes (*p* < 0.0001) (Table 2). 

## 4. Discussion

In human cases of neurotrophic keratopathy due to surgery for intracranial lesions, which can eliminate the effects of microbial or surgical manipulation, we found that affected eyes had lower nerve fiber density, increased dendritic cells, and larger corneal epithelial basal cells. Although pathological changes by repeated corneal erosion may have affected the corneal findings, our manuscript reveals aberrant corneal homeostasis in neurosurgery-induced neurotrophic keratopathy.

Corresponding to the decrease in corneal perception using the Cochet–Bonnet corneal anesthesiometer, the density of the corneal trigeminal nerve was significantly lower in the affected eye than in the contralateral eye. This was supported by the IVCM findings. The decrease in nerve fiber density in Case 3 (3198 μm/mm^2^), which had a relatively short post-onset period, was lower than that in Cases 1 and 2 (1032 μm/mm^2^, 1600 μm/mm^2^, respectively), both of whom had a longer history of trigeminal neuropathy. This indicates that long-term neuropathy leads to morphological changes in nerves over time. Since there was no history of epithelial erosion in Case 3, the effects of neuropathy may be influenced by the duration of neuropathy.

Along with the decrease in nerve fiber density, the degree of tortuosity was significantly altered in neurotrophic eyes. The degree of tortuosity of the trigeminal nerve in severe neurotrophic corneas may have been evident because of the degradation of nerve fibers. In recent years, corneal nerve regeneration has been reported when minimally invasive corneal neurotization is performed within 6 months of intracranial surgery for trigeminal neurotrophic keratitis. Although the efficacy of this procedure in long-term neurotrophic keratitis such as our cases is not yet known, it is a very promising new treatment [20].

The concept of neural-immune coupling has been established, and neural regulation is known to govern the behavior of immune cells. In the present study, dendritic cells in the neurotrophic eyes increased in Cases 1, 2, and 3. This was supported by animal models by Hamrah et al. [10], in which dendritic cells were increased in the neurotrophic cornea. They also described that the dendritic cells, which are normally in the corneal limbus, easily expand to the central cornea in a denervated cornea. This phenomenon might suggest that a denervated cornea easily reacts to stimuli and is prone to inflammation. We could not clarify the changes in the distribution of dendritic cells due to the narrow measurement range of the confocal laser microscope. However, excessive corneal opacities observed on OCT may be due to excessive inflammation induced in the neurotrophic cornea.

In this study, we observed an increase in the size of basal cells in the cornea under neurotrophic conditions. This increase in epithelial basal cell area or decrease in density may be due to decreased tropism, abnormal differentiation, or expression of compensation due to insufficient cell supply from stem cells. Liu et al. also stated that herpes-induced corneal stem cell exhaustion affects basal cell density in the central cornea [21]. In our opinion, the same phenomenon was observed in our patients with neurotrophic keratitis. It has been reported that corneal basal cells are tightly associated with neural axons through the fusion of cell membranes [22,23]. In addition, according to Okada and colleagues, the trigeminal nerve influences the differentiation of limbal stem cells [24]. Although the exact mechanism underlying the increase in the basal cell area remains unclear, it is possible that impairment of the trigeminal nerve may have affected the division and differentiation of basal cells, resulting in changes in their size.

In our observation, the mean area of epithelial basal cells was 259 ± 86.5 μm^2^ in the neurotrophic eye and 185 ± 45.9 μm^2^ in the normal eye. This means that the mean cell densities were 4346 ± 1451 cells/mm^2^ and 5760 ± 1429 cells/mm^2^, respectively. Previously, Liu et al. and Sterenczak et al. reported epithelial basal cell densities of normal cornea were 3650 ± 746 cells/mm^2^ and 8190.39 cells/mm^2^ [21,25]. Although there is a wide range of values for basal cell density based on IVCM observation, our data do not deviate from their data [21,25]. The high-intensity reflections in the epithelial surface layer after wound healing observed in OCT might be due to hyperplasia caused by abnormal cell differentiation or dysplasia caused by insufficient intercellular adhesion system.

It is also possible that the tear quantity and/or quality over the cornea might have had an impact on the ocular surface health status. Eyes with corneal neuropathy with a Schirmer test value of less than or equal to 5mm appear to have several erosion recurrences. Schirmer test results of the contralateral eyes in all cases were 3-8mm, which are low for a normal eye. The neurotrophic condition of the other eye might have some effects on the contralateral eyes. Future prospective studies in a large number of subjects would provide interesting information in our belief.

Tight junctions might be related to these findings because reports in the field of dermatology have shown that tight junctions in the skin epithelium are disrupted by neuropathy [26]. However, this requires further research.

This study has several limitations. First, the number of cases was small, and the age of cases was variable. Second, we performed IVCM observation at one timepoint for each patient, and changes over time could not be analyzed. A neurosurgery-induced neurotrophic keratopathy is a rare condition, Therefore, five images of IVCM were used for analysis in each patient. Especially, for basal cells, the area of 15 cells per image was measured. In other words, we measured the area of 75 cells per patient. Although this is small size samples, the statistical process is accurate. Our findings are the results at this time, and we think further studies with a larger number of cases are needed.

In conclusion, clinical cases of neurotrophic keratopathy caused by surgery of intracranial diseases has rarity value. We believe that our findings will help to clarify the pathogenesis of neurotrophic keratopathy.

## 5. Conclusions

We found a decrease in nerve fibers and an increase in dendritic cells in neurosurgery-induced neurotrophic keratopathy, as well as in an animal model [10,11,12,13,27]. The increase in the size of basal cells in the neurotrophic cornea is a new finding. These changes may be related to the clinical characteristics of excessive inflammation, epithelial detachment, irregular wound healing, and severe scars.

## Figures and Tables

**Figure 1 jcm-11-03804-f001:**
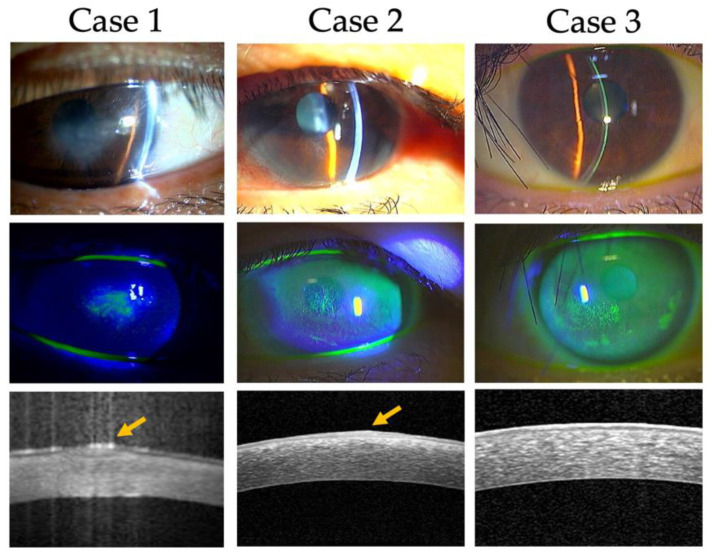
Characteristics of a neurotrophic keratopathy evaluated by slit-lamp examination. Neurotrophic keratopathy is present in the paracentral region with superficial stromal opacity. The vortex pattern, known as the normal healing pattern, was not observed with fluorescein staining. Focal thickening of the epithelium was sustained after erosion, which indicates disturbances in the healing process. Anterior segment optical coherence tomography also showed focal thickening of the epithelium with high density.

**Figure 2 jcm-11-03804-f002:**
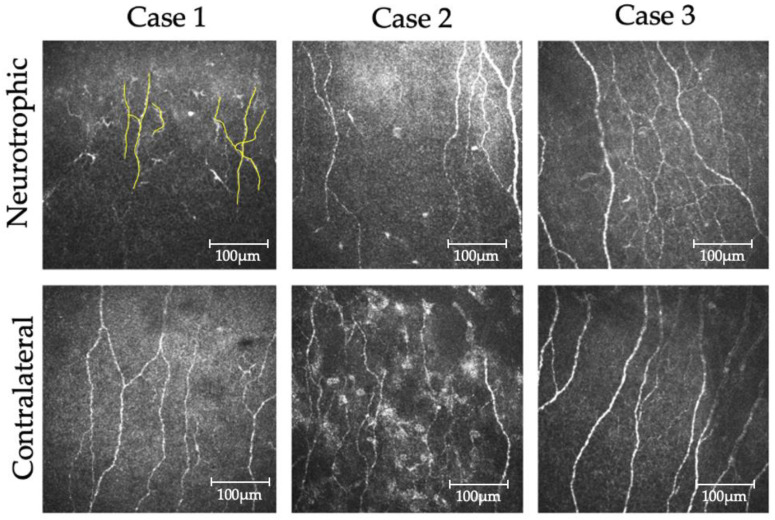
Sub-basal nerve plexus assessed by in vivo confocal microscopy. The upper panels show the neurotrophic cornea, and the lower panels show the contralateral cornea. We traced all nerve fibers (yellow line) and determined the mean density of the sub-basal nerve plexus using five photographs in each patient. Note that there was a significant decrease in nerve fiber density in the neurotrophic eye.

**Figure 3 jcm-11-03804-f003:**
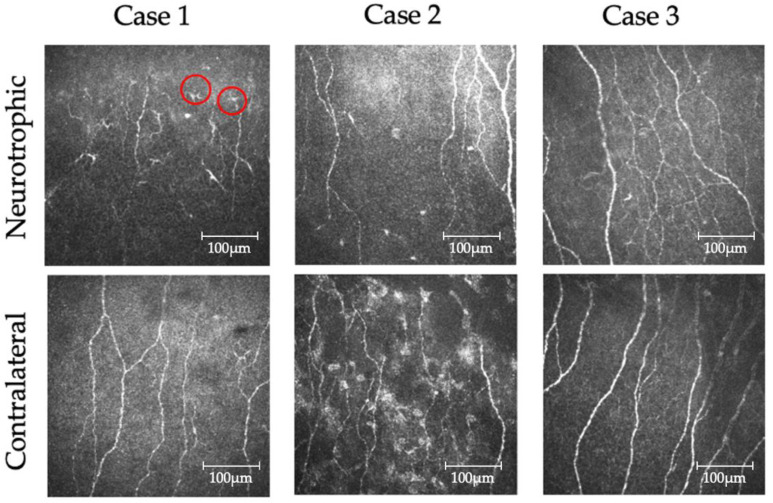
In vitro confocal microscopy observation of dendritic cells. All dendritic cells (red circle) were manually counted. The numbers of dendritic cells were counted from 5 photographs in each patient. The mean number of dendritic cells in the neurotrophic eyes was larger than that in the contralateral eyes.

**Figure 4 jcm-11-03804-f004:**
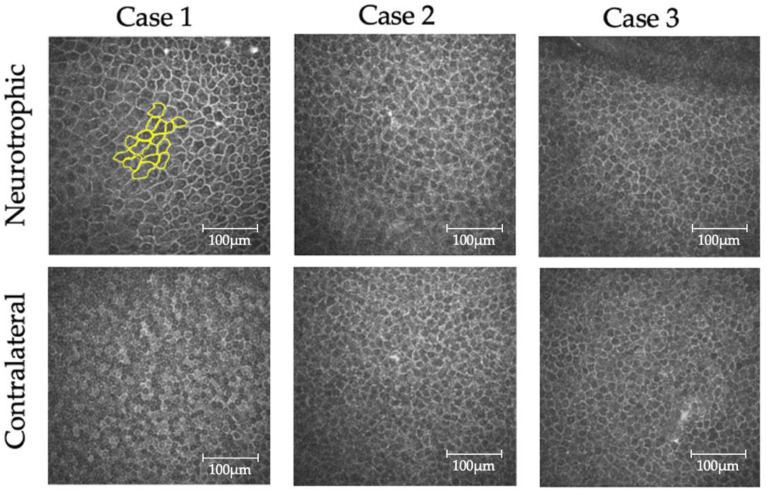
Observation of basal cells by in vivo confocal microscopy. Fifteen cells per photo were randomly selected and marked, and the mean cell size was determined from five photographs in each patient using Image J. Note the larger size of the basal cells in neurotrophic corneas than in contralateral corneas.

**Table 1 jcm-11-03804-t001:** Patients’ background and clinical findings.

	Case 1	Case 2	Case 3
Age/sex	78/F	79/F	52/F
Neurotrophic side	Right	Left	Right
Primary illness	auditory schwannoma	auditory schwannoma	γ-knife brain tumor
Postoperative periods (years)	22	30	2
Corneal sensation (mm)	<5	<5	<5
Schirmer test I (mm) N: Neurotrophic eye C: Contralateral eye	N: 5 C: 3	N: 5 C: 8	N: 5 C: 8
Recurrence of corneal erosion	Several times (once a year)	Several times (once a year)	No erosion
Mackie’s classification	1–2	1–2	1

**Table 2 jcm-11-03804-t002:** Comparison of both corneas by IVCM (analysis of variance: ANOVA).

	Nerve Fiber Density (μm/mm^2^)		Number of Dendritic Cells (Cells/mm^2^)		Size of Basal Epithelial Cell (μm^2^)	
	Neurotrophic	Contralateral	Neurotrophic	Contralateral	Neurotrophic	Contralateral
Case 1	1032 ± 459.5	1890 ± 426.3	7.50 ± 6.10	10.0 ± 6.10	227 ± 87.0	157 ± 27.0
Case 2	1600 ± 445.4	1904 ± 417.6	27.5 ± 12.2	10.0 ± 3.10	327 ± 82.7	192 ± 49.1
Case 3	3198 ± 435.7	2932 ± 322.7	10.0 ± 3.10	1.25 ± 2.50	226 ± 38.6	209 ± 42.5
mean	1943 ± 1000 **	2242 ± 600.3	30.8 ± 21.8 ##	6.25 ± 5.59	259 ± 86.5 ##	185 ± 45.9
*P* value	** *P* = 0.0347		## *P* < 0.0000		## *P* < 0.0000	

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
