# Peer review of "Aberrant Corneal Homeostasis in Neurosurgery-Induced Neurotrophic Keratopathy"

_jcm, 2022, doi:10.3390/jcm11133804_

Round 1

Reviewer 1 Report

Oba and colleagues investigated the in vivo confocal microscopy (IVCM) findings of neurotrophic keratopathy (NK) due to neurosurgical procedures in 3 patients and compared the affected eye with the contralateral eye. Despite the small sample size, there are interesting findings of the dendritic cells and the basal epithelial cells that have not been reported previously. Although the manuscript reads well, I have a few major and some minor concerns as mentioned below.

General comments:

- The authors have included 4 cases, however, OCT and IVCM are not available in one patient. Since clinical features of surgery-induced NK have been described before, I do not think that the 4th case would add to the manuscript.

- The authors have compared the affected eye with the contralateral healthy eye. I recommend removing “non-comparative” throughout the manuscript, as the fellow eye is the comparator.

Major specific comments:

- Line 15: “neurotrophic keratopathy induced by intracranial lesions”; this is different from neurosurgery-induced NK. I suppose the authors meant “induced by surgery for intracranial lesions” rather than “induced by intracranial lesion” itself.

- Line 16 and lines Line 58–59: “8 eyes of 4 patients”; please clarify in the abstract that all cases were unilateral NK and fellow eyes were used as control.

- Line 54: “pure trigeminal neuropathy”; the disease duration was 22 and 30 years in two cases. How do the authors prove that previous infections and/or multiple recurrent erosions do not contribute to the findings?

- Methods: please add the type and brand of OCT and IVCM.

- Methods: please explain how the regions of interest for the analysis of IVCM images were chosen? Were the measurements performed on the corneal center?

- Methods: were the graders masked to the eye?

- Table 1: laterality of the affected eye should be added.

- Table 1: Schirmer test results in the contralateral eyes (C) are 1–8 mm, which is too low for a normal eye. Please check and correct.

- Results: Were there concurrent facial or other cranial nerve palsies in any of the cases?

- Line 248: “as well as in an animal model”; does this refer to a previous study? If so, please cite.

Line 173–174: “which can eliminate the effects of microbial or surgical manipulation”; again, how do the authors prove that previous infections and/or multiple recurrent erosions do not contribute to the findings in patients with a prolonged postoperative interval?

Line 182: paralysis refers to pathology in the efferent fibres and inability to use muscle, I suggest using “trigeminal neuropathy” instead of “paralysis” throughout the manuscript.

- Discussion: please also include the findings of the recently published paper that describes IVCM findings in 12 cases in comparison with 12 controls (PMID: 35179100).

Minor specific comments:

Line 20: either mention AS-OCT in the methods (perhaps line 15 after “(IVCM) findings”) or delete OCT findings from the results.

Lines 58–59 and lines 67–68: please clarify that only 4 eyes had neurotrophic keratopathy.

Figures: please add scale bar to the Figures.

Line 179–181: please provide individual data in the results section, preferably in Table 2.

Line 192–196: please add reference (Hamrah et al.)

Author Response

Thank you for your meaningful review. We will attach the file of response.

We appreciate to have your review again.

With best regards Shimpei Oba

Reviewer 2 Report

You present an interesting case series of 4 patients suffering from neurotrophic keratopathy (NK). Please find my comments and requests below:

1. Introduction - the main diagnostic criterium of NK is hypaesthesie or anaesthesie of the cornea. Please include this information in the first paragraph. What is your hypothesis? Did you expect more dendritic cells or bigger basal cells? What were the endpoints of this "prospective study"?

2. In line 31: "cannot regenerate properly" rather than are damaged

3. Line 46: "have been modified"

4. Materials and methods: in my opinion this is a case series rather than a prospective  study. A prospective study requieres a description of: in- and exclusion criteria, recruitment  strategy, planned case number according to a statistical analysis. I miss all these elemnts in your paper.

5. Classification of NK nach Mackie should be added in the Methods section

6. Methods: model, firma and country of IVCM shoul be added

7. Methods: were the examiners (ICVM) blinded regarding affected  and unaffected eye? IVCM is a kind of subjective examination. The unblinded examiner can find images that match to the expected results. Please provide this information and explain the examination procedure in the methods section.

8. Line 73: what exactly means "5 best images"? best quality? most cells?

9. Line 75: Instead trigeminal nerve density: subbasal nerve plexus

10. Please provide NK stage (Mackie) to your patients (ex. in table 1)

11. whre was the corneal sensitivity tested? centrally or in quadrants? or in the lesion? I am not sure if you can give 0 mm as a result of Cochet-Bonnet - it would mean you touched the cornea not with the "suture".

12. Please provide us with the therapy of the patients (past and at the time of the study)

13. Schirmer I or II?

14. I think that you cannot speak of significancy and p-value in case of 6 eyes...  Please provide a statistical expertise answering this question.

Author Response

(The authors gave the same response as above.)

Round 2

Reviewer 1 Report

The authors did very well with the revision and have addressed all of my comments in the revised manuscript. I have no further comments.

Reviewer 2 Report

I agree to your corrections. The manuscript may be accepted in this form.